# Post-Disaster Resilience Optimization for Road–Bridge Transportation Systems Considering Economic Loss

**Jiangbin Zhao** [1,2], **Mengtao Liang** [1,2], **Zaoyan Zhang** [1,2], **Xiangang Cao** [1,2], **Qi Lu** [1,2] and **Zhiqiang Cai** [3,*]

1   School of Mechanical Engineering, Xi'an University of Science and Technology, Xi'an 710054, China; zhaojiangbin@xust.edu.cn (J.Z.); liangmengtao@stu.xust.edu.cn (M.L.); zhangzaoyan@stu.xust.edu.cn (Z.Z.); xgcao_xust@163.com (X.C.); luqi@xust.edu.cn (Q.L.)

2   Shaanxi Key Laboratory of Mine Electromechanical Equipment Intelligent Detection and Control, Xi'an 710054, China

3   Department of Industrial Engineering, School of Mechanical Engineering, Northwestern Polytechnical University, Xi'an 710072, China

*   Correspondence: caizhiqiang@nwpu.edu.cn

**Abstract:** After a disaster, the recovery sequence of damaged bridges in a road–bridge transportation system greatly influences system restoration time and total economic loss. In this paper, the skew of recovery trajectory is introduced to evaluate the average restoration time, and the total economic loss is extended to consider the indirect loss, such as the energy consumption of detours or the emergency service fee. So, the post-disaster resilience optimization model is constructed by minimizing the total economic loss. The improved genetic algorithm is developed to obtain the optimal recovery scheme for damaged bridges by considering the recovery sequence and repair modes. The composition and influence factors of total economic loss are analyzed through three experiments. The experimental results show that the indirect loss accounts for approximately half of the economic loss, while the higher price of emergency service promotes the reduction of indirect loss using the expedited modes to repair damaged bridges. Moreover, to minimize the total economic loss, it is essential to design the optimal recovery scheme (repair sequence and repair mode) wisely to balance the conflicts between indirect loss and direct loss.

**Keywords:** post-disaster restoration; recovery time; resilience optimization; total economic loss; road–bridge transportation system

## 1. Introduction

A road–bridge transportation system is made up of $N$ nodes and $M$ arcs. Nodes represent major road intersections, commercial hubs, or key destinations in a community, while arcs represent road sections with bridges. This transportation system is susceptible to adverse weather conditions like floods and hurricanes. Post-disaster refers to the situation, activity, process, or phase that occurs after a natural or man-made disaster. Therefore, the post-disaster resilience optimization of road–bridge networks is significant for helping governments to restore the normal operation of transportation systems in a cost-effective manner.

The use of evaluation methods of system resilience in the recovery stage is important for analyzing the performance of infrastructure after a disaster. In this section, community resilience evaluation methods previously used to analyze the performance of infrastructure systems after a natural hazard are summarized [1,2]. A basic framework has been used to assess the system resilience of urban transportation systems to reduce the consequences of disruptions [3]. The resilience of civil infrastructure was assessed based on three criteria, including system reliability, redundancy, and resilience [4]. Recovery time-sensitive network resilience has been developed by considering bridge closures to schedule recovery scenarios and reduce recovery time [5]. Analyzing the resilience of the Maritime Silk Road

shipping network by combining route data and transmission data has provided a scientific basis for ensuring structural resilience after unexpected disasters [6]. The resilience of a transportation system after external shocks has been assessed in order to explore optimal network configurations and better restore system performance [7]. Also, establishing a new reliability framework helped to make informed decisions for the construction and operation of the Sichuan–Tibet Railway [8]. Resilience has become a major issue in preventing and resolving the risk of large-scale power outages in cyber–physical power systems [9]. Communication network reliability indicators should be promoted and supported with reference to the development of power networks [10]. The existing research is more focused on analyzing system performance in terms of system resilience, but seldom do such studies introduce recovery capability to evaluate the system resilience. Therefore, it is essential to construct a recovery ability-based resilience index to assess system performance effectively.

A reasonable recovery scheme of transportation systems can restore a system's resilience in a cost-effective way. Transportation network resilience refers to the ability of a system to recover normal function after natural or man-made disasters, which affect the structure, function, management, and environment of the system [11]. Different types of transportation network resilience evaluation methods have been analyzed, including mathematical programming, heuristic algorithms, simulation models, network analysis, system dynamics, etc. [12]. Also, a decision framework was constructed to prioritize bridge repair after a disruptive event using three network performance measures, including the functional measure, topological measure, and social measure [13]. This paper explores a new pathway towards achieving the seismic resilience of road networks under earthquake hazards, which was developed by leveraging post-shock rapid responses to minimize functionality losses [14]. A structured framework has been established to plan the recovery scheme of a the transportation network after a disaster [15]. Additionally, a robust performance measure was introduced to evaluate the resilience of transportation networks under seismic conditions, which was used to determine optimal post-hazard bridge recovery strategies [16]. An average time delay was proposed to evaluate the performance of the road–bridge transportation system after an earthquake, and an optimization model of post-earthquake recovery scheduling for bridges was constructed to determine the optimal recovery scheme [17]. The resilience of congested urban road networks after earthquakes has been assessed using topological network functional indicators for obtaining the optimal resource planning for the recovery stage [18]. A multistage stochastic program with decision-dependent uncertainty was proposed to jointly determine post-disaster inspection and restoration activity schedules to minimize roadway downtime and maximize the probability of completing the repairs successfully during the recovery period [19]. To maximize the resilience of the grid after a disaster, it is necessary to prioritize post-disaster maintenance by calculating the resilience measures of grid components and their corresponding importance measures [20]. A stratified sampling approach of utility data has been used to assess and rank the impact of interdependencies between power systems and other infrastructure on power system resilience [21]. A systematic review of the human factors affecting the post-disaster resilience modeling of infrastructure systems has also been conducted [22]. In summary, the current recovery optimization methods focus on mathematical programming, simulation models, or heuristic methods to obtain an optimal recovery strategy. Therefore, it is an exploratory way of solving the recovery optimization problem using intelligent swarm algorithms.

Through a comparative analysis of post-disaster recovery in urban and rural communities, it was found that the efficiency and quality of post-disaster recovery are affected by resource constraints such as budget, manpower, and materials [23]. Economic factors have an important impact on enhancing a system's ability to recover. The tremendous financial and societal losses caused by the disruption of highway bridges have been analyzed by considering long-term resilience under multiple natural hazards [24]. To solve the recovery ability optimization problem, economic loss has been analyzed by considering indirect losses and maintenance costs [25]. Specifically, the resilience of Queensferry Crossing in the

UK was increased by considering the cost of potential mitigating measures concerning its closure [26]. A continuous-time Markov decision process framework based on life-cycle cost has been proposed to determine the optimal earthquake-resistant levels of rebuilt bridges by considering the social and economic consequences [27]. Spontaneously triggered risk has also been evaluated by considering the subregional hazard, socioeconomic exposure, and triggered risk from the traffic jam or interruption caused by road inundation [28]. To increase the resilience of transport networks, cost-effective monitoring and maintenance strategies have been implemented by considering the socioeconomic impacts of disruptions due to bridge failures or closures [29]. To evaluate the accessibility of the optimization plan for post-disaster road network restoration, direct costs such as restoration costs and indirect costs such as service interruption losses and environmental pollution losses have been considered [30]. Considering post-disaster restoration costs and indirect costs such as personnel and property losses, two types of resilience improvement measures were determined before and after disasters, and a multiobjective optimization model based on a genetic algorithm was established to maximize the expected resilience value of infrastructure systems [31]. In order to measure and optimize the recovery ability of the post-disaster traffic system, that is, the ability of the system to recover normal functions after disaster, an elastic optimization model based on budget and travel time constraints was proposed to determine an optimal post-disaster traffic system recovery plan by considering direct economic loss, environmental damage, personnel and property loss, and other indirect losses after disaster maintenance and management [32]. The research gap which exists concerning economic loss analysis during system recovery is summarized in Table 1. Most of the existing research works in the literature focus on maintenance loss and property loss, but team management loss, energy consumption, and emergency service are largely ignored in terms of indirect economic loss. To some extent, the indirect economic loss caused by emergency services or energy consumption may cause considerable economic loss compared with maintenance cost. More engineering teams mean that the restoration of a road–bridge system can be sped up. Therefore, it is essential to consider the team management loss, energy consumption, and emergency services.

**Table 1.** The research gap of the economic loss analysis during system recovery.

| References | Direct Economic Loss | | | Indirect Economic Loss | | |
|:---:|:---:|:---:|:---:|:---:|:---:|:---:|
| | Maintenance Loss | Team Management Loss | Energy Consumption | Emergency Services | Environmental Pollution | Property Loss |
| [23] | ✓ | ✗ | ✓ | ✗ | ✗ | ✗ |
| [24] | ✓ | ✓ | ✗ | ✗ | ✓ | ✓ |
| [25] | ✓ | ✗ | ✓ | ✗ | ✓ | ✓ |
| [26] | ✓ | ✗ | ✗ | ✗ | ✗ | ✓ |
| [27] | ✓ | ✗ | ✓ | ✓ | ✓ | ✗ |
| [28] | ✓ | ✗ | ✓ | ✗ | ✗ | ✓ |
| [29] | ✓ | ✗ | ✓ | ✗ | ✓ | ✗ |
| [30] | ✓ | ✗ | ✗ | ✗ | ✗ | ✓ |
| [31] | ✓ | ✓ | ✗ | ✗ | ✓ | ✓ |

✓—yes; ✗—no.

After the comprehensive consideration of the reported research work, the motivation of this paper became to fully consider economic loss in order to improve the resilience optimization model of the road–bridge transportation system after a disaster. Economic loss includes maintenance costs, team management costs, and indirect costs. To minimize the overall economic loss and effectively manage the recovery time of road and bridge networks, an optimal recovery strategy is obtained using a genetic algorithm and elite strategy. Therefore, the research work in this paper can better address post-disaster challenges and enhance the system's recovery capabilities. The innovations of this paper are as follows.

(1) The resilience index is constructed by considering the independent pathways between two nodes to better evaluate the recovery capability of a road–bridge network. (2) This paper evaluates the indirect economic loss by considering the energy consumption and emergency services for the post-disaster recovery of a road–bridge network. (3) Due to the the limited recovery time, multiple engineering teams with different repair modes are introduced to speed up the recovery process.

The remainder of the paper is organized as follows. Section 2 establishes the post-disaster resilience optimization model under the required restoration time. Section 3 introduces the procedures of the improved genetic algorithm. Section 4 analyzes the composition and influence factors of total economic loss through three experiments. Finally, Section 5 summarizes the findings of this paper.

## 2. Description of the Post-Disaster Resilience Optimization Problem

In this paper, a road–bridge transportation system is constructed which consists of nodes (commercial hub) and arcs (bridges with different lengths of roads); it is an independent system because the failure of one bridge will not result in the failure of other bridges. The length of roads and the average daily traffic of each bridge are known, and the recovery time and unit cost of different modes (expedited mode and normal mode) can be provided by engineers who can measure the damage level of bridges after a disaster. The probability of repairing the damaged bridge to its normal state depends on the level of damage and the bridge's normal state, which is equal to the ratio of the difference between these two state levels to the normal state level. Once the repair modes and maintenance sequence are determined, the total recovery time can be evaluated by arranging different engineering teams until all engineering teams complete all the maintenance tasks. Increased numbers of engineering teams can speed up the bridge's restoration, but adding an engineering team results in an extra management fee.

During the restoration process, the total economic loss includes indirect loss which occurs due to energy consumption and emergency services needed to connect a damaged bridge, the direct loss involves the repairing of all of the damaged bridges, and the team management loss involves the managing of the engineering teams. The post-disaster resilience optimization problem involves the determination of the restoration scheme needed to minimize the total economic loss under the constraints of the average recovery time, which is a single-period planning problem.

### 2.1. Assumptions

To clearly describe the post-disaster resilience optimization problem, the assumptions are summarized as follows.

- The road–bridge transportation system is a typical structure with known information about nodes and arcs, such as the length of all arcs and the average daily traffic of each arc.
- After a disaster, the damage level for all bridges can be determined by an inspection, and the recovery time of damaged bridges under different modes is related to the damage level and the skill experiences of engineering teams, which are generated by a random simulation.
- Bridges can be regarded as multistate components; a bridge with a lower state means that it is in a better situation.
- Once the damaged bridges are determined, the available engineering teams can prioritize the maintenance of damaged bridges with a quicker recovery time, which can improve the effective performance of the system to an extent.

### 2.2. Average Recovery Time

System resilience is an important indicator for evaluating the system's ability to resist degradation or recover to its normal state. To analyze the resilience of the road–bridge transportation system clearly when recovery sequence $x$ and repair modes $y$ are known,

$R(t; x, y)$ is an index of system performance which considers the reliability and redundancy of arcs, the criticality of nodes, and the importance level of arcs. $R(t; x, y)$ can be evaluated using Equation (1) as follows.

$$R(t; x, y) = \sum_{i=1}^{N} w_i / (N - 1) \sum_{j=1, j \neq i}^{N} \sum_{k=0}^{K_{(i,j)}} w_k(i, j) \prod_{\forall a \in p_k(i,j)} (1 - q_a / s_a) \tag{1}$$

where $x$ is the sequence square with $n^2$ elements, and all of the elements are either 0 or 1; $y$ is a column matrix with $n$ elements, representing the repair modes of $n$ damaged bridges; $w_i$ is the criticality of node $i$ ($1 \leq i \leq N$), which depends on the distance of the nearest emergency response facilities and is the criticality node with the higher weight, considering the nearest emergency response facilities; $K(i, j)$ is the number of independent pathways between node $i$ and node $j$, which can be regarded as the redundancy of arcs; independent pathways include the paths connecting the nodes without any crossover with other paths between these two nodes [33]; $w_k(i, j)$ is the importance level of the arcs in the $k$-th independent pathway, which is the relative importance between the pathway length and its average daily traffic; $q_a$ is the damage level of the arc $a$; and $s_a$ represents the level of the arc $a$.

Considering the probability of repairing the damaged bridges, the skew of the recovery trajectory (SRT) is used to evaluate the average recovery time of the road–bridge transportation system by integrating $s(x, y)$ [34]. The infinitesimal method is used to evaluate $s(x, y)$ approximately, though it may take a long time to evaluate the SRT by calculating the sum of each piece, so SRT can be calculated using Equation (2).

$$s(x, y) = \frac{\int_0^T R(t; x, y) t \, dt}{\int_0^T R(t; x, y) \, dt} \approx \frac{\sum_{l=0}^{k} (l \Delta t) R(l \Delta t; x, y) \Delta t}{\sum_{l=0}^{k} R(l \Delta t; x, y) \Delta t} \tag{2}$$

where $T$ is the evaluation time point, which can be divided into $k$ pieces with the same length, so the length of each piece is $\Delta t = \frac{T}{t}$.

### 2.3. Evaluation of the Total Economic Loss

Total economic loss consists of maintenance loss, management loss, and indirect loss. The detailed calculation of different types of loss is introduced as follows.

### 2.3.1. Maintenance Loss $L_M$

Maintenance loss consists of repairing the damaged bridges in expedited mode or normal mode, which can be calculated via Equation (3).

$$L_M = \sum_{i=1}^{n} \left( (1 - y_i) t_{D_i} \mathbf{C}_{1(\sum_{j=1}^{n} j x_{ij})} + y_i t_{D_i} \mathbf{C}_{2(\sum_{j=1}^{n} j x_{ij})} \right)$$
$$t_{D_i} = (1 - y_i) \mathbf{T}_{1(\sum_{j=1}^{n} j x_{ij})} + y_i \mathbf{T}_{2(\sum_{j=1}^{n} j x_{ij})} \tag{3}$$

where $t_{D_i}$ is the repair time of the damaged bridge $i$ when the repair mode is determined; $\mathbf{T}_1$ and $\mathbf{T}_2$, row matrixes with $n$ elements, are the recovery time with the expedited mode or the normal mode of all damaged bridges, respectively; $\mathbf{T}_{1(i)}$ represents the recovery time of the $i$-th damaged bridge under the expedited mode, and $\mathbf{T}_{2(i)}$ represents the recovery time of the $i$-th damaged bridge under the normal mode; $x_{ij}$ is the element of $x$, which means that damaged bridge $i$ is assigned to the $j$-th maintenance activity; $y_i$ is a symbol of the repair mode for the damaged bridge $i$, where $y_i = 1$ represents that restoring the damaged bridge $i$ uses the normal mode, while $y_i = 0$ represents the expedited mode; $\mathbf{C}_1$ and $\mathbf{C}_2$, row matrixes with $n$ elements, are the unit repair cost of all damaged bridges with the expedited mode or the normal mode, respectively; $\mathbf{C}_{1(i)}$ represents the repair cost of the $i$-th damaged bridge under the expedited mode, and $\mathbf{C}_{2(i)}$ represents the repair cost of the $i$-th damaged bridge under the normal mode.

### 2.3.2. Team Management Loss

The team management loss consists of the management fee of engineering teams, which is in proportion to the number of engineering teams. The management fee of m engineering teams can be calculated using Equation (4).

$$L_C = mC_3 \tag{4}$$

where $m$ is the number of engineering teams, and $C_3$ is the management fee when one engineering team is assigned to the restoration process.

### 2.3.3. Indirect Loss

Indirect loss depends on whether the shortest pathway between two nodes exists. If two nodes of a damaged bridge exist on a pathway after a disaster, the energy cost of a detour needs to be considered, which relates to average daily traffic, the length of the alternative shortest pathway, and the recovery time. If there are no pathways between two nodes, which means that the bridge has failed, emergency service can be used to deliver vehicles through the damaged bridges.

However, the recovery sequence and repair mode can impact the completion time of each damaged bridge, which has a close relationship with indirect loss. When the recovery sequence $x$ and repair mode $y$ are known, knowing how to arrange the tasks for $m$ engineering teams is critical. We consider using the minimal completion time to assign repair tasks to engineering teams as follows.

1.  Select the engineering team with the minimal completion time based on $t_{D_i}$ to repair the $i$-th damaged bridge;
2.  For the remaining $n-1$ damaged bridges, implement the following process:
    *   Assign the repair task of the $i$-th bridge to all the engineering teams, and choose the engineering team with the quickest repair task to finish the repair of damaged bridge $i$;
    *   If multiple engineering teams have the same total task time, the idle engineering team takes this task as a priority;
    *   For the next damaged bridge, perform $i = i + 1$;
    *   When all the damaged bridges have been repaired, stop the task assignment.
3.  According to the assignment, determine $T_0 = \{t_1, \cdots, t_i, \cdots, t_n\}$ and $B_0 = \{b_1, \cdots, b_i, \cdots, b_n\}$. $T_0$ is the time when the repairing process of the damaged bridges is completed according to $x$ and $y$; $B_0$ corresponds to $T_0$, which records the bridge sequence based on the completion time.

The energy cost of a detour refers to the additional costs faced when extra fuel is consumed during the restoration process, which is equal to the sum of the difference in fuel costs after and before the recovery of all damaged bridges. The energy cost can be evaluated using Equation (5).

$$L_{I_1} = \sum_{i=1}^{n} (t_i - t_{i-1}) M_i^{ADT} (P(b_i, t_i)(L(b_i, t_i) - L_0(b_i))C_4) \tag{5}$$

where $P(b_i, t_i) = 1$ represents that there exists at least one shortest pathway between two nodes of bridge $b_i$ at time $t_i$; otherwise $P(b_i, t_i) = 0$; $t_i$ represents the repair completion time for the $i$-th damaged bridge; $M_i^{ADT}$ denotes the average daily traffic on the $i$-th damaged bridge; $L(b_i, t_i)$ is the length of the shortest path between the nodes of bridge $b_i$ when repairing the damaged bridge at time $t_i$; while $L_0(b_i)$ is the length of the shortest path between the nodes of bridge $b_i$ when all the bridges are repaired; and $C_4$ is the unit price per vehicle traveling one kilometer.

Emergency services are involved in the service cost of transferring a vehicle from one end of the bridge to the other as a temporary measure when the bridge collapses. The

emergency fee depends on the service time and average daily traffic between two nodes of a bridge, so it can be calculated via Equation (6).

$$L_{I_2} = \sum_{i=1}^{n} (t_i - t_{i-1}) M_i^{ADT} (1 - P(b_i, t_i)) C_5 \tag{6}$$

where $C_5$ is the unit price of the emergency delivery service transporting the vehicles cross a damaged bridge.

*2.4. Post-Disaster Resilience Optimization Model under the Required SRT*

The purpose of this problem is to determine the recovery scheme under the required average recovery time for minimizing total economic loss. The objective function is to minimize the total economic loss, which is shown in Equation (7). The decision variables are the recovery sequence $x$ and the repair modes $y$ of $n$ damaged bridges. The constraints include the recovery sequence (Equation (8)), the repair modes (Equation (9)), the target bridge damage level after repair (Equation (10)), the number of simultaneous engineering teams (Equation (11)), and the required recovery time (Equation (12)). Therefore, the post-disaster resilience optimization model under the required SRT is shown as follows.

$$\min C(x, y) = L_M + L_C + L_{I_1} + L_{I_2} \tag{7}$$

$$s.t. \ \sum_{i=1}^{n} x_{ij} = 1, \ \sum_{j=1}^{n} x_{ij} = 1, x_{ij} = \{0, 1\} \tag{8}$$

$$y_i = \begin{cases} 0, \text{if bridge } i \text{ is repaired in expedited mode} \\ 1, \text{if bridge } i \text{ is repaired in normal mode} \end{cases} \tag{9}$$

$$q_{b_i}^{(t)} = \begin{cases} 0, \ t \geq t_i \\ q_{b_i}, t < t_i \end{cases}, \forall t \in [0, T] \tag{10}$$

$$\sum_{b=1}^{n} [t \geq t_i - t_{D_{b_i}}] \cdot [t \leq t_i] \leq m, \forall t \in [0, T] \tag{11}$$

$$s(x, y) \leq T_{\max} \tag{12}$$

where Equation (8) means that only one repair sequence can be selected for a bridge and only one bridge can be selected for each repair; Equation (9) defines the repair mode of bridge $i$; and Equation (10) limits the damage level of bridge $i$ at time point $t$, and bridge $i$ is restored to its normal state when $t \geq t_i$, in which $q_{b_i}$ is the damage level of bridge $i$. Equation (11) requires that the number of working engineering teams is no more than $m$. Equation (12) represents the limitation of the average recovery time, where $T_{\max}$ is the required average recovery time.

## 3. Procedures of the Solving Algorithm

An algorithm with a simple principle and high convergence speed is important to solve the complex optimization problem outlined in this paper. The SRT and total economic loss depend on the recovery sequence and the repair modes of damaged bridges, and there are $2^n n!$ potential recovery schemes once the damaged bridges are determined. Moreover, the structure of the road–bridge system is a complex network, meaning that it always takes a long time to evaluate the SRT. To solve this complex optimization problem effectively, it is essential to select an algorithm with simple principle and high convergence speed. The genetic algorithm (GA) has good global search ability and can search for all the solutions in the solution space quickly, without falling into the local optimal solutions [35]. The GA starts from the population, has potential parallelism, can carry out a simultaneous comparison of multiple individuals, and improve the search efficiency and robustness. The elitism strategy of the GA, replacing the worst solution with the best solution, is helpful to obtain an optimal recovery scheme for the recovery optimization of a road–

bridge network [36]. Therefore, we use the standard GA to solve the nonlinear constrained optimization problem. The procedure of the improved GA is shown in Figure 1, including the population initialization, decode method, selection, crossover, mutation, and elitist strategy.

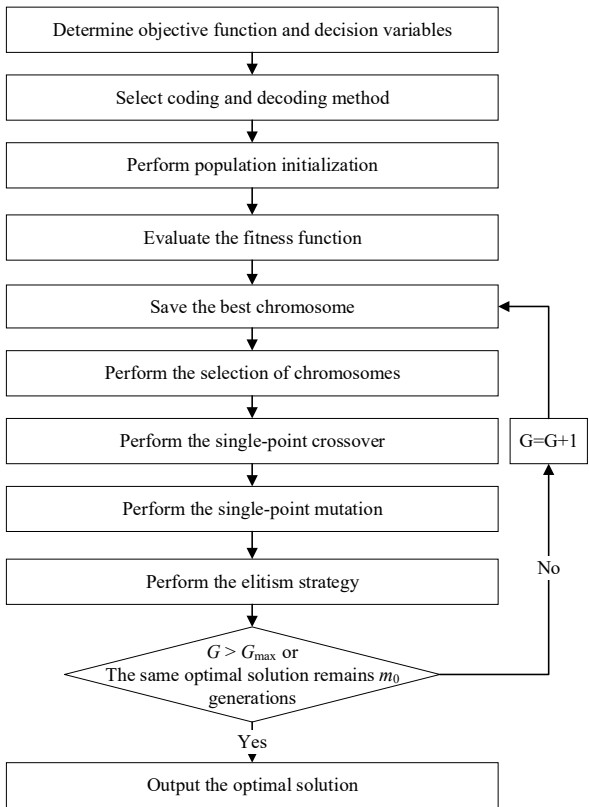

**Figure 1.** The procedures of the improved GA.

To solve this post-disaster resilience optimization model, we modified the coding and decoding method of individuals, the crossover operation, the elitism strategy, and termination conditions because the decision variables are discrete.

### 3.1. The Coding and Decoding Method of Individuals

We used the real number code method to express the recovery scheme, which can be represented by a row matrix with $2n$ elements, including the recovery sequence (the first $n$ elements on the left) and the repair mode (the last $n$ elements on the right). The recovery sequence is a permutation of the damaged bridges; the normal mode is noted as 1, and the expedited mode is noted as 0. There is an example in Figure 2 which describes the coding method clearly. As can be seen in Figure 2, the recovery sequence is 3–6–4–2–5–1, and the damaged bridges 1, 2, and 6 should use the expedited repair mode while bridges 3, 4, and 5 should use the normal repair mode.

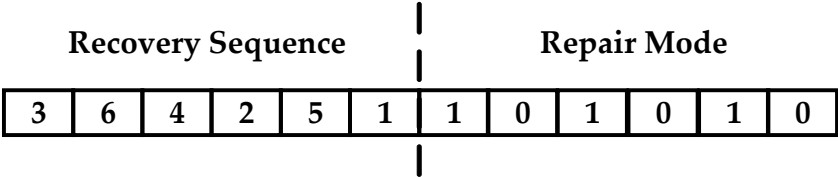

**Figure 2.** An example of the coding method for an individual.

The decoding method should be implemented by the task assignment with the minimal completion time outlined in Section 2.3 when the number of engineering teams is known.

If the repair time of six damaged bridges is known, $t_D$ = {0.9, 1.2, 0.7, 0.5, 0.4, 0.8}, for which the unit is a month. According to the procedures, the decoding result is shown in Figure 3. So, we can obtain $B_0$ = {3, 6, 4, 2, 5, 1} and $T_0$ = {0.7, 0.8, 1.2, 1.2, 1.6, 1.7}.

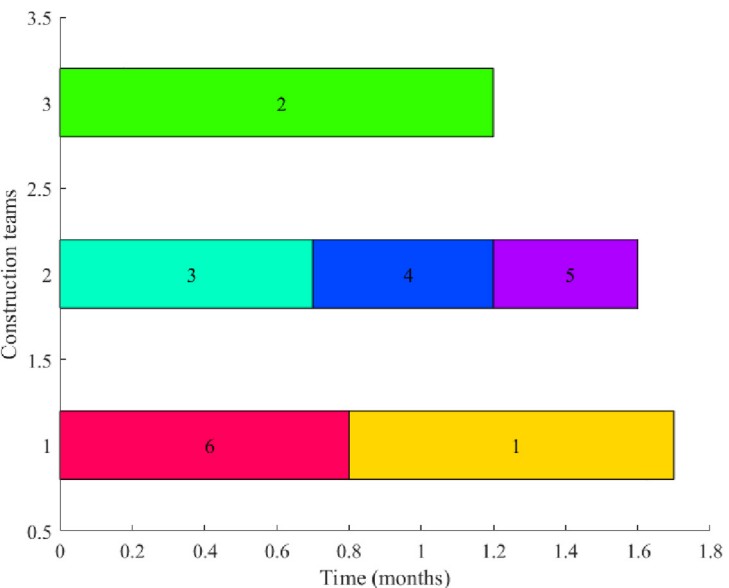

**Figure 3.** An example of the decoding method for an individual.

### 3.2. Crossover Operation

There may be repetitive genes in an individual in the recovery sequence after the single-point crossover, but the repair mode does not exist in this conflict. We need to adjust the repetitive genes by the gene not appearing in the current individual so that all the genes in the recovery sequence part are unique. An example of adjusting the process in the crossover is shown in Figure 4. We find that for individual 1, the repetitive genes (genes 2 and 5) are at positions 1 and 3, so the repetitive genes are replaced by genes 1 and 3, respectively. Similarly, for individual 2, the repetitive genes at positions 1 and 2 should be replaced by genes 2 and 5, respectively. The positions within the grey shadow mean that the genes have been adjusted to the valid genes.

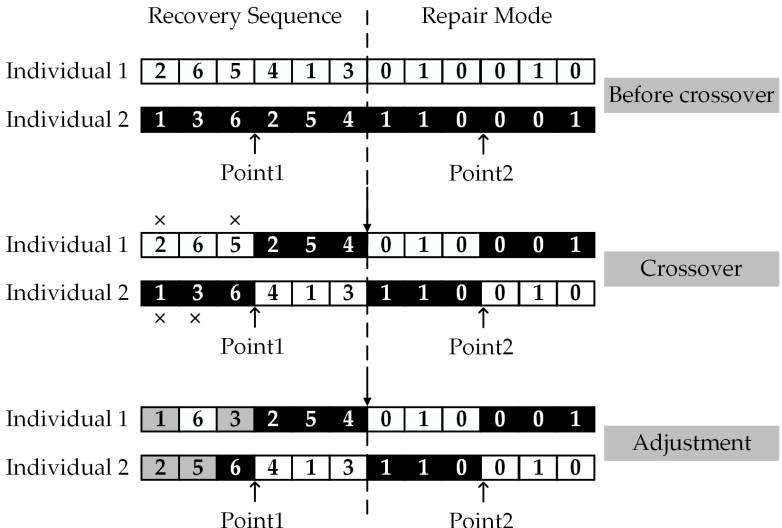

× represents the conflicted genes in the individuals after crossover.

**Figure 4.** An example of the crossover.

### 3.3. Elitism Strategy

Currently, the elitist strategy means that the individual with the lowest fitness level should be replaced by the individual with the highest fitness level. The advantage of this strategy is that it can improve the convergence speed and stability and avoid falling into local optimum [37]. Based on this subprocess, the best individual will be utilized, meaning that the results will improve with more generations.

### 3.4. Termination Conditions

There are two termination conditions of the improved GA, and achieving one of them will terminate the proposed algorithm. The first termination condition is when the current generation is larger than the maximum generation; the other condition is when the algorithm stops once the optimal solution remains unchanged for $m_0$ generations.

### 4. Simulation Experiments for Analyzing the Total Economic Loss

A hypothetical road–bridge transportation system with 30 nodes and 37 arcs is shown in Figure 5 [34,38]. The nodes are commercial hubs, which are important areas but hard to destroy. An arc consists of a road and a bridge that can easily collapse after a disaster. As seen in Figure 5, nodes 9 and 17 represent the emergency response facilities, which can provide maintenance resources. Before the disaster, the length of each arc and the average daily traffic (ADT) are known and available. After the disaster, engineers assess the damage level, the unit price, and restoration time using different repair modes. Bridges are considered to be damaged when the damage level is above zero, which implies the need to restore the bridge using the normal repair mode or the expedited repair mode. The parameters of the bridges in the networks are listed in Table 2.

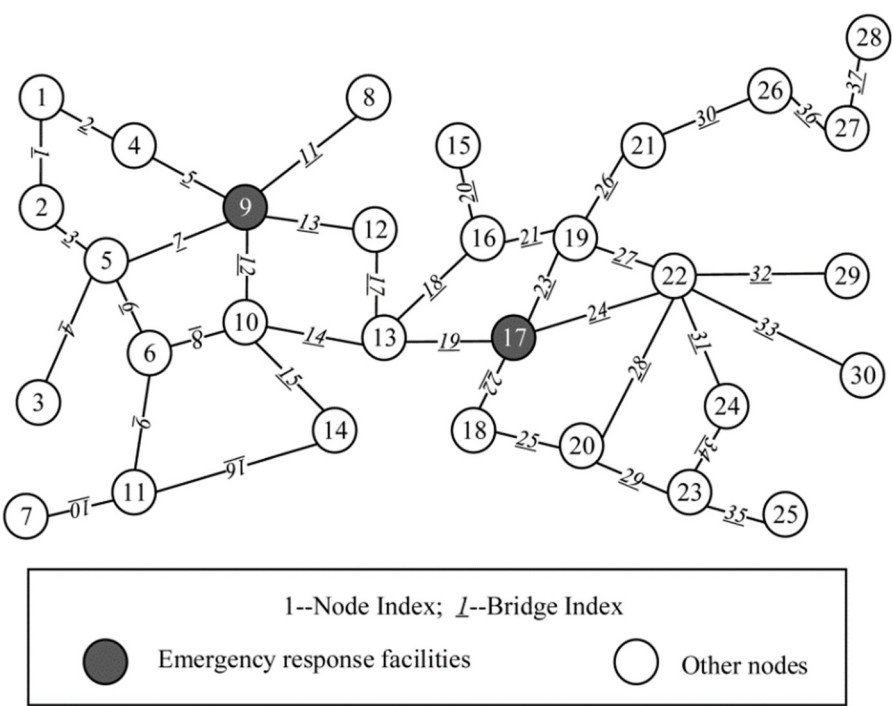

**Figure 5.** Hypothetical road–bridge transportation system.

**Table 2.** The parameters in the road–bridge transportation system.

| Bridge ID | Node 1 | Node 2 | ADT (vehicle/day) | Length of Arcs (km) | Damage Level | $C_1$ (million dollars/month) | $C_2$ (million dollars/month) | $T_1$ (months) | $T_2$ (months) |
|---|---|---|---|---|---|---|---|---|---|
| 1 | 1 | 2 | 2200 | 22 | 1.71 | 3.98 | 8.61 | 4.10 | 2.14 |
| 2 | 1 | 4 | 1900 | 18 | 0.88 | 3.43 | 7.31 | 2.49 | 1.64 |
| 3 | 2 | 5 | 2000 | 15 | 2.79 | 4.87 | 9.75 | 9.04 | 4.77 |
| 4 | 3 | 5 | 1500 | 36 | 0.00 | 0.00 | 0.00 | 0.00 | 0.00 |
| 5 | 4 | 9 | 1900 | 27 | 2.51 | 4.5 | 9.52 | 5.25 | 3.45 |
| 6 | 5 | 6 | 2200 | 20 | 0.00 | 0.00 | 0.00 | 0.00 | 0.00 |
| 7 | 5 | 9 | 700 | 30 | 0.00 | 0.00 | 0.00 | 0.00 | 0.00 |
| 8 | 6 | 10 | 2400 | 20 | 0.00 | 0.00 | 0.00 | 0.00 | 0.00 |
| 9 | 6 | 11 | 2600 | 30 | 1.16 | 3.55 | 7.57 | 3.32 | 2.00 |
| 10 | 7 | 11 | 300 | 20 | 2.56 | 4.66 | 9.61 | 6.52 | 3.47 |
| 11 | 8 | 9 | 800 | 38 | 0.84 | 3.33 | 6.48 | 2.38 | 1.37 |
| 12 | 9 | 10 | 900 | 24 | 0.84 | 3.25 | 6.36 | 1.71 | 1.19 |
| 13 | 9 | 12 | 2500 | 26 | 0.00 | 0.00 | 0.00 | 0.00 | 0.00 |
| 14 | 10 | 13 | 600 | 27 | 0.79 | 2.61 | 5.52 | 1.44 | 0.88 |
| 15 | 10 | 14 | 2000 | 28 | 0.79 | 2.88 | 6.03 | 1.65 | 1.09 |
| 16 | 11 | 14 | 500 | 46 | 0.14 | 1 | 1.50 | 0.55 | 0.45 |
| 17 | 12 | 13 | 2500 | 23 | 1.17 | 3.57 | 7.58 | 3.34 | 2.01 |
| 18 | 13 | 16 | 2800 | 32 | 0.00 | 0.00 | 0.00 | 0.00 | 0.00 |
| 19 | 13 | 17 | 1300 | 24 | 0.70 | 2.53 | 4.91 | 1.40 | 0.75 |
| 20 | 15 | 16 | 1700 | 18 | 0.00 | 0.00 | 0.00 | 0.00 | 0.00 |
| 21 | 16 | 19 | 1500 | 16 | 2.58 | 4.82 | 9.71 | 6.65 | 4.59 |
| 22 | 17 | 18 | 1200 | 15 | 1.79 | 4.34 | 8.99 | 4.75 | 2.94 |
| 23 | 17 | 19 | 1500 | 23 | 0.00 | 0.00 | 0.00 | 0.00 | 0.00 |
| 24 | 17 | 22 | 700 | 37 | 0.03 | 0.75 | 1.00 | 0.25 | 0.15 |
| 25 | 18 | 20 | 1800 | 16 | 0.00 | 0.00 | 0.00 | 0.00 | 0.00 |
| 26 | 19 | 21 | 900 | 23 | 2.25 | 4.46 | 9.32 | 5.02 | 3.30 |
| 27 | 19 | 22 | 600 | 19 | 1.19 | 3.82 | 7.68 | 3.99 | 2.09 |
| 28 | 20 | 22 | 800 | 46 | 3.86 | 5.29 | 10.16 | 10.21 | 6.10 |
| 29 | 20 | 23 | 1400 | 22 | 0.00 | 0.00 | 0.00 | 0.00 | 0.00 |
| 30 | 21 | 26 | 2800 | 28 | 0.01 | 0.6 | 0.80 | 0.15 | 0.10 |
| 31 | 22 | 24 | 1900 | 31 | 0.29 | 1.2 | 1.80 | 0.60 | 0.48 |
| 32 | 22 | 29 | 2900 | 35 | 0.00 | 0.00 | 0.00 | 0.00 | 0.00 |
| 33 | 22 | 30 | 1300 | 47 | 0.87 | 3.4 | 7.15 | 2.42 | 1.44 |
| 34 | 23 | 24 | 900 | 17 | 0.00 | 0.00 | 0.00 | 0.00 | 0.00 |
| 35 | 23 | 25 | 2200 | 18 | 0.00 | 0.00 | 0.00 | 0.00 | 0.00 |
| 36 | 26 | 27 | 700 | 18 | 0.00 | 0.00 | 0.00 | 0.00 | 0.00 |
| 37 | 27 | 28 | 3000 | 16 | 0.00 | 0.00 | 0.00 | 0.00 | 0.00 |

As can be seen in ref. [15], we have analyzed the numbers of engineering teams which have had an important influence on the SRT and total economic loss without considering the management fee of engineering teams. Moreover, we also found that indirect loss accounts for a relatively large portion of total economic loss. In this paper, we aim to study the following three problems in detail via three experiments: (1) composition analysis of total economic losses; (2) the optimal number of engineering teams; (3) the impact of the unit price of emergency services on the restoration sequence.

### 4.1. Design of Three Experiments

In Experiment 1, the mentioned road–bridge transportation system with a different number of damaged bridges ($n$ = 5, 10, 15, and 20) and engineering teams ($m$ = 1, 2, and 3) is implemented to analyze the composition of the total economic loss for the optimal recovery scheme. Experiment 1 focuses on the change in total economic loss and the percentage of maintenance loss, management loss, and indirect loss as system scales or engineering teams increase. All the damaged bridges are generated randomly and the damage levels after the disaster are listed in Table 1. Moreover, the system parameters are set as follows.

The unit management fee of engineering teams $C_3$ is USD 100 million; the average price of fuel consumption $C_4$ is 0.1 USD/km/vehicle; and the average price of emergency service $C_5$ is 1 dollar/vehicle.

Experiment 2 involves the impact of the number of engineering teams and the team management loss of each engineering team on the total economic loss when there are 20 damaged bridges after a disaster. Experiment 2 records the changes in the total economic loss as the increase in engineering teams ($m$ = 1, 2, 3, 4, 5, 6, 7, 8, 9, and 10) with a different management loss for each team ($C_3$ = 100, 200, and 300). All the system parameters are the same as in Experiment 1 except for $C_3$.

Considering that the unit price of fuel $C_4$ always remains unchanged, Experiment 3 is implemented to analyze the impact of the unit price of emergency service on the total economic loss and restoration sequence. We select different unit prices, $C_5$ = 0.25, 0.5, 0.75, 1, 1.25, 1.5, 1.75, 2, 2.25, 2.5, 2.75, 3, 3.25, 3.5, 3.75, 4, 4.25, 4.5, 4.75, 5, 5.25, 5.5, 5.75, and 6 USD/vehicle.

The algorithm parameters of the three experiments are the same and are listed as follows. The population size *popsize* = 100, $G_{max}$ = 100, $r_0$ = 50, crossover probability $p_c$ = 0.9, and mutation probability $p_m$ = 0.1.

### 4.2. Result Analysis of Three Experiments

#### 4.2.1. Result of Experiment 1

The results of Experiment 1 are listed in Table 3, and the case ($m$, $n$) represents the case when the number of engineering teams is m and the number of damaged bridges is $n$. In Table 2, the maintenance loss, management loss, and indirect loss are listed in detail. All the units of economic losses are shown as millions of USD.

**Table 3.** The economic loss of different cases with different *m* and *n*.

| *m* | *n* | Damaged Bridges | Indirect Loss | Maintenance Loss | Team Management Loss | Total Economic Loss |
|---|---|---|---|---|---|---|
| 1 | 5 | [1, 5, 16, 21, 27] | 4.24 | 1.73 | 1 | 6.97 |
| 2 | 5 | [1, 5, 16, 21, 27] | 2.52 | 1.73 | 2 | 6.25 |
| 3 | 5 | [1, 5, 16, 21, 27] | 2.1 | 1.73 | 3 | 6.83 |
| 1 | 10 | [3, 5, 10, 11, 14, 16, 21, 22, 24, 28] | 8.67 | 3.67 | 1 | 13.34 |
| 2 | 10 | [3, 5, 10, 11, 14, 16, 21, 22, 24, 28] | 5.08 | 3.51 | 2 | 10.59 |
| 3 | 10 | [3, 5, 10, 11, 14, 16, 21, 22, 24, 28] | 4.61 | 3.52 | 3 | 11.13 |
| 1 | 15 | [3, 5, 9, 10, 11, 14, 15, 16, 21, 22, 24, 26, 27, 28, 30] | 10.47 | 4.67 | 1 | 16.14 |
| 2 | 15 | [3, 5, 9, 10, 11, 14, 15, 16, 21, 22, 24, 26, 27, 28, 30] | 5.8 | 4.67 | 2 | 12.47 |
| 3 | 15 | [3, 5, 9, 10, 11, 14, 15, 16, 21, 22, 24, 26, 27, 28, 30] | 4.93 | 4.37 | 3 | 12.3 |
| 1 | 20 | [1, 2, 3, 5, 9, 10, 11, 12, 14, 16, 17, 19, 21, 22, 24, 26, 27, 28, 30, 33] | 13.49 | 5.33 | 1 | 19.82 |
| 2 | 20 | [1, 2, 3, 5, 9, 10, 11, 12, 14, 16, 17, 19, 21, 22, 24, 26, 27, 28, 30, 33] | 7.52 | 5.3 | 2 | 14.82 |
| 3 | 20 | [1, 2, 3, 5, 9, 10, 11, 12, 14, 16, 17, 19, 21, 22, 24, 26, 27, 28, 30, 33] | 5.94 | 5.03 | 3 | 13.97 |

To better analyze the composition of total economic loss, the changes in economic loss in different cases are shown in Figure 6. As can be seen from Figure 6, the economic loss becomes higher with the increase in damaged bridges. Once the system scale is determined, the indirect loss decreases with the increase in $m$, and the team management loss increases with the increase in $m$, while the maintenance losses using different engineering teams are similar. Adding engineering teams can decrease indirect loss quickly, which can better decrease economic loss when the number of damaged bridges is higher.

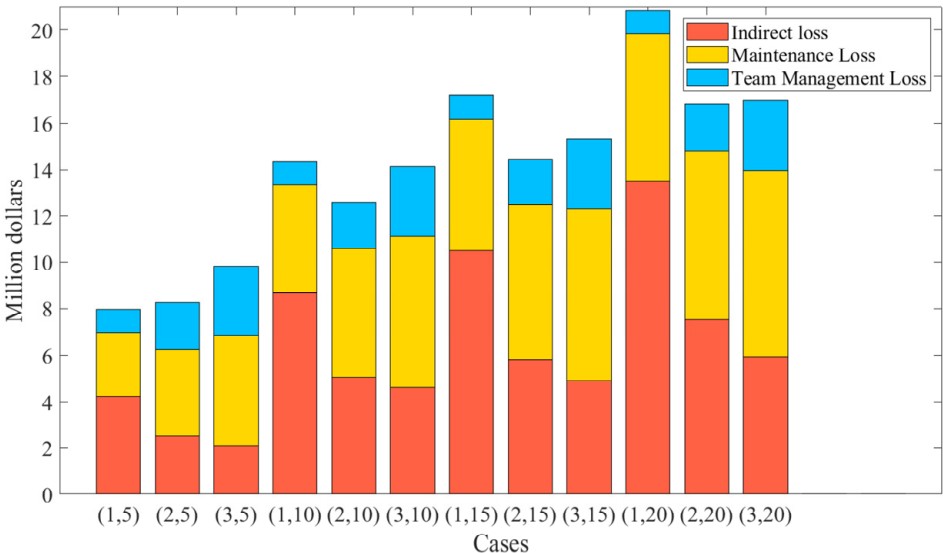

**Figure 6.** The composition of economic loss for 12 cases.

Moreover, the percentage of economic loss can be calculated, and the percentage of three losses is shown in Figure 7. The percentage of management loss increases as *m* increases when the damaged bridges are the same, while the percentage of indirect loss becomes lower with the increase in engineering teams. The maintenance loss accounts for about 30% in these 12 cases. For example, the indirect loss accounts for 60% when *m* = 1, while it decreases to about 40% when *m* = 3.

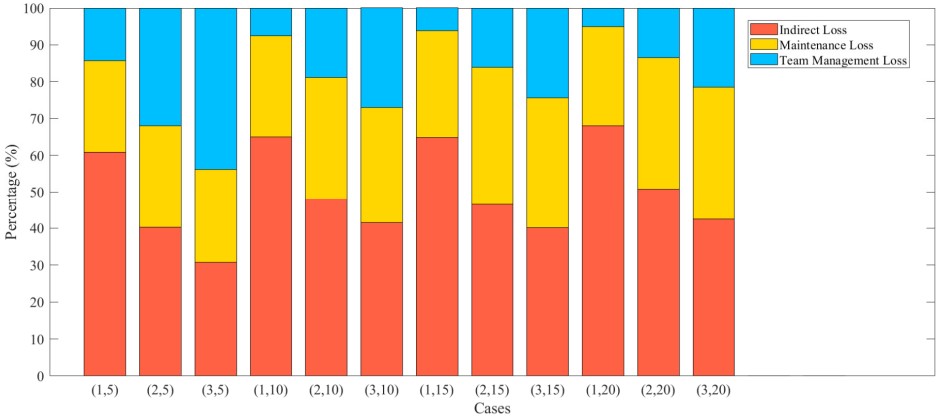

**Figure 7.** The percentage of economic loss for 12 cases.

By analyzing the composition of economic loss, we find that the number of engineering teams has an important influence on total economic loss, and increasing the number of engineering teams properly can decrease total economic loss effectively.

### 4.2.2. Result of Experiment 2

For Experiment 2, total economic loss and its composition with the optimal recovery scheme are summarized in Table 4. According to the results, changes in the economic loss are shown in Figure 8.

**Table 4.** The changes in economic loss with different $m$ and $C_3$.

| $m$ | $C_3 = 100$ | | | | $C_3 = 200$ | | | | $C_3 = 300$ | | | |
|---|---|---|---|---|---|---|---|---|---|---|---|---|
| | TEL | TML | ML | IL | TEL | TML | ML | IL | TEL | TML | ML | IL |
| 1 | 19.82 | 1 | 5.33 | 13.49 | 21.69 | 2 | 5.33 | 14.36 | 22.66 | 3 | 5.33 | 14.33 |
| 2 | 14.82 | 2 | 5.30 | 7.52 | 16.88 | 4 | 5.33 | 7.55 | 18.52 | 6 | 5.33 | 7.19 |
| 3 | 13.96 | 3 | 5.03 | 5.94 | 17.06 | 6 | 5.03 | 6.04 | 19.55 | 9 | 5.17 | 5.38 |
| 4 | 13.69 | 4 | 5.03 | 4.66 | 18.02 | 8 | 5.17 | 4.84 | 22.56 | 12 | 4.93 | 5.63 |
| 5 | 14.10 | 5 | 4.92 | 4.18 | 19.37 | 10 | 5.03 | 4.35 | 23.89 | 15 | 5.03 | 3.86 |
| 6 | 15.38 | 6 | 4.63 | 4.74 | 20.68 | 12 | 5.04 | 3.64 | 27.07 | 18 | 5.18 | 3.89 |
| 7 | 15.63 | 7 | 5.04 | 3.59 | 22.90 | 14 | 5.04 | 3.86 | 29.64 | 21 | 4.84 | 3.80 |
| 8 | 16.02 | 8 | 5.04 | 2.98 | 24.34 | 16 | 5.04 | 3.30 | 32.66 | 24 | 4.96 | 3.70 |
| 9 | 17.11 | 9 | 4.92 | 3.18 | 26.24 | 18 | 5.19 | 3.05 | 35.27 | 27 | 5.04 | 3.23 |
| 10 | 17.99 | 10 | 4.96 | 3.03 | 28.30 | 20 | 4.87 | 3.43 | 38.13 | 30 | 5.19 | 2.95 |

TEL: total economic loss; TML: team management loss; ML: maintenance loss; IL: indirect loss.

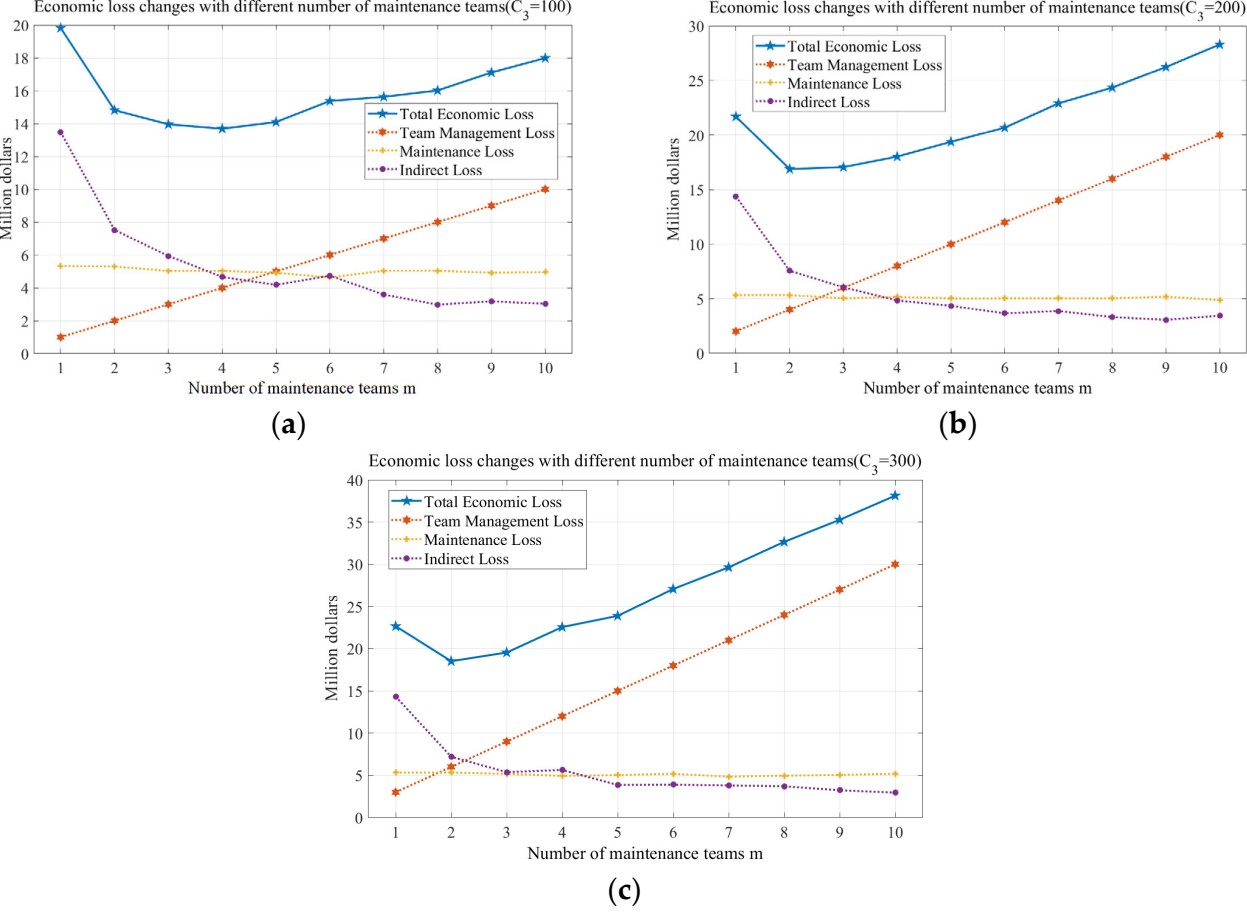

**Figure 8.** Changes in economic loss with different numbers of engineering teams. (**a**) $C_3 = 100$; (**b**) $C_3 = 200$; (**c**) $C_3 = 300$.

For Figure 8, the maintenance losses under different $C_3$ remain essentially constant with the increase in the number of engineering teams. Because these three situations have the same damaged bridges, the maintenance loss is almost USD 5 million. Indirect losses decrease rapidly with the number of engineering teams, while fixed loss increases linearly with the number of engineering teams. The total loss decreases and then increases with the increase in the engineering teams, showing a V-shaped trend. The lowest economic loss appears when $m = 4$, $m = 2$, and $m = 2$ for $C_3 = 100$, 200, and 300, respectively. With the increase in $C_3$, the lowest economic loss appears earlier. Therefore, the management fee has

a large impact on economic loss, so the optimal number of engineering teams should be determined by balancing the conflicts between management loss and indirect loss. If the management fee is lower, we can use more engineering teams.

### 4.2.3. Result of Experiment 3

The results of Experiment 3 are listed in Table 5, where we can find economic loss, maintenance loss, SRT, recovery scheme, and the number of normal modes. Twenty damaged bridges and their repair information are detailed; the SRT and number of normal modes decrease with the increase in the price of emergency services, while EL and ML increase as $C_5$ increases. The first row in the recovery scheme represents the recovery sequence number of these 20 damaged bridges, and all the elements are integers from 1 to 20; while the second row is the repair modes. With the increase in $C_5$, we find that element '1' appears less and less. Particularly, almost all the damaged bridges use expedited mode to be restored when $C_5 >$ USD 3.5, which means all the damaged bridges should be restored quickly when the emergency service fee is higher. So, it is easy to understand that the maintenance loss increases and the SRT decreases with $C_5$. However, the economic loss increases quickly as $C_5$ increases, which means that a higher emergency fee promotes engineering teams to complete maintenance tasks as soon as possible.

**Table 5.** Changes in the recovery scheme as $C_5$ increases.

| $C_5$ | SRT | TEL | ML | Recovery Scheme | | | | | | | | | | | | | | | | | | | | Number of Normal Modes |
|---|---|---|---|---|---|---|---|---|---|---|---|---|---|---|---|---|---|---|---|---|---|---|---|---|
| 0.25 | 16.3 | 1141.7 | 845.8 | 13 | 19 | 5 | 8 | 18 | 4 | 14 | 16 | 15 | 17 | 3 | 10 | 20 | 11 | 2 | 12 | 9 | 6 | 7 | 1 | 8 |
| | | | | 1 | 0 | 1 | 0 | 0 | 0 | 0 | 1 | 0 | 0 | 1 | 0 | 0 | 0 | 0 | 0 | 1 | 1 | 1 | 1 | |
| 0.5 | 14.0 | 1170.8 | 877.3 | 5 | 8 | 2 | 19 | 20 | 10 | 13 | 14 | 17 | 11 | 3 | 9 | 1 | 15 | 7 | 4 | 18 | 16 | 6 | 12 | 5 |
| | | | | 0 | 0 | 0 | 1 | 0 | 1 | 1 | 0 | 0 | 0 | 0 | 0 | 0 | 0 | 0 | 0 | 0 | 1 | 1 | 0 | |
| 0.75 | 14.4 | 1346.8 | 901.5 | 5 | 8 | 11 | 19 | 7 | 14 | 10 | 18 | 17 | 15 | 13 | 20 | 9 | 3 | 2 | 12 | 16 | 4 | 6 | 1 | 6 |
| | | | | 0 | 0 | 1 | 0 | 0 | 0 | 1 | 0 | 0 | 0 | 0 | 0 | 1 | 0 | 1 | 0 | 1 | 0 | 1 | 0 | |
| 1 | 14.9 | 1365.8 | 903.7 | 5 | 8 | 2 | 19 | 10 | 20 | 13 | 1 | 11 | 3 | 4 | 7 | 9 | 12 | 14 | 17 | 6 | 18 | 15 | 16 | 4 |
| | | | | 0 | 0 | 0 | 0 | 1 | 0 | 0 | 0 | 0 | 0 | 0 | 0 | 0 | 1 | 0 | 1 | 0 | 1 | 0 | 1 | |
| 1.25 | 14.6 | 1481.5 | 893.6 | 5 | 8 | 2 | 19 | 11 | 9 | 1 | 13 | 7 | 14 | 15 | 17 | 3 | 10 | 20 | 12 | 4 | 18 | 16 | 6 | 6 |
| | | | | 0 | 0 | 0 | 0 | 0 | 1 | 1 | 0 | 1 | 0 | 1 | 0 | 0 | 0 | 0 | 0 | 0 | 0 | 1 | 1 | |
| 1.5 | 14.0 | 1490.9 | 917.4 | 5 | 8 | 2 | 19 | 20 | 13 | 1 | 11 | 3 | 14 | 9 | 4 | 12 | 15 | 6 | 7 | 10 | 17 | 16 | 18 | 1 |
| | | | | 0 | 0 | 0 | 0 | 0 | 0 | 0 | 0 | 0 | 0 | 0 | 0 | 0 | 0 | 1 | 0 | 0 | 0 | 0 | 0 | |
| 1.75 | 14.2 | 1584.2 | 910.1 | 2 | 8 | 5 | 19 | 3 | 7 | 1 | 20 | 10 | 4 | 11 | 14 | 13 | 6 | 15 | 9 | 17 | 16 | 18 | 12 | 2 |
| | | | | 0 | 0 | 0 | 0 | 0 | 0 | 0 | 0 | 0 | 0 | 0 | 0 | 0 | 1 | 0 | 0 | 0 | 0 | 0 | 0 | |
| 2 | 15.1 | 1667.8 | 917.4 | 2 | 8 | 5 | 19 | 3 | 7 | 1 | 20 | 10 | 4 | 11 | 14 | 13 | 6 | 15 | 9 | 17 | 16 | 18 | 12 | 1 |
| | | | | 0 | 0 | 0 | 0 | 0 | 0 | 0 | 0 | 0 | 0 | 0 | 0 | 0 | 1 | 0 | 0 | 0 | 0 | 0 | 0 | |
| 2.25 | 15.1 | 1704.2 | 918.6 | 5 | 8 | 20 | 19 | 11 | 2 | 3 | 7 | 1 | 4 | 9 | 13 | 10 | 14 | 6 | 12 | 16 | 15 | 17 | 18 | 2 |
| | | | | 0 | 0 | 0 | 0 | 0 | 0 | 0 | 0 | 0 | 0 | 0 | 0 | 0 | 0 | 1 | 1 | 0 | 0 | 0 | 0 | |
| 2.5 | 15.2 | 1788.2 | 917.5 | 11 | 20 | 2 | 19 | 1 | 5 | 8 | 16 | 3 | 9 | 7 | 10 | 13 | 4 | 14 | 12 | 15 | 6 | 17 | 18 | 2 |
| | | | | 0 | 0 | 0 | 0 | 0 | 0 | 0 | 0 | 0 | 0 | 0 | 0 | 0 | 0 | 0 | 0 | 1 | 1 | 0 | 0 | |
| 2.75 | 14.3 | 1905.7 | 917.4 | 5 | 7 | 19 | 13 | 2 | 14 | 8 | 11 | 3 | 4 | 15 | 9 | 12 | 1 | 20 | 10 | 17 | 6 | 16 | 18 | 1 |
| | | | | 0 | 0 | 0 | 0 | 0 | 0 | 0 | 0 | 0 | 0 | 0 | 0 | 0 | 0 | 0 | 0 | 0 | 1 | 0 | 0 | |
| 3 | 15.0 | 1906.3 | 917.4 | 5 | 8 | 2 | 19 | 11 | 20 | 14 | 1 | 9 | 7 | 13 | 3 | 15 | 10 | 16 | 4 | 12 | 18 | 6 | 17 | 1 |
| | | | | 0 | 0 | 0 | 0 | 0 | 0 | 0 | 0 | 0 | 0 | 0 | 0 | 0 | 0 | 0 | 0 | 0 | 0 | 1 | 0 | |
| 3.25 | 14.3 | 2000.0 | 917.4 | 5 | 8 | 2 | 19 | 20 | 3 | 11 | 1 | 4 | 9 | 7 | 14 | 13 | 6 | 12 | 15 | 10 | 17 | 16 | 18 | 1 |
| | | | | 0 | 0 | 0 | 0 | 0 | 0 | 0 | 0 | 0 | 0 | 0 | 0 | 0 | 1 | 0 | 0 | 0 | 0 | 0 | 0 | |
| 3.5 | 14.5 | 2041.0 | 933.0 | 5 | 8 | 2 | 19 | 11 | 1 | 12 | 20 | 13 | 3 | 9 | 10 | 7 | 14 | 4 | 15 | 18 | 16 | 6 | 17 | 0 |
| | | | | 0 | 0 | 0 | 0 | 0 | 0 | 0 | 0 | 0 | 0 | 0 | 0 | 0 | 0 | 0 | 0 | 0 | 0 | 0 | 0 | |
| 3.75 | 14.8 | 2157.7 | 917.4 | 5 | 8 | 14 | 20 | 1 | 19 | 2 | 7 | 11 | 3 | 15 | 10 | 13 | 4 | 12 | 9 | 16 | 17 | 6 | 18 | 1 |
| | | | | 0 | 0 | 0 | 0 | 0 | 0 | 0 | 0 | 0 | 0 | 0 | 0 | 0 | 0 | 0 | 0 | 0 | 0 | 1 | 0 | |
| 4 | 14.8 | 2183.4 | 933.0 | 5 | 8 | 2 | 19 | 20 | 1 | 12 | 10 | 11 | 7 | 3 | 13 | 9 | 4 | 14 | 16 | 15 | 18 | 17 | 6 | 0 |
| | | | | 0 | 0 | 0 | 0 | 0 | 0 | 0 | 0 | 0 | 0 | 0 | 0 | 0 | 0 | 0 | 0 | 0 | 0 | 0 | 0 | |
| 4.25 | 14.4 | 2219.3 | 933.0 | 12 | 8 | 11 | 19 | 20 | 2 | 5 | 7 | 1 | 9 | 3 | 10 | 14 | 13 | 4 | 15 | 17 | 16 | 6 | 18 | 0 |
| | | | | 0 | 0 | 0 | 0 | 0 | 0 | 0 | 0 | 0 | 0 | 0 | 0 | 0 | 0 | 0 | 0 | 0 | 0 | 0 | 0 | |
| 4.5 | 14.3 | 2310.8 | 933.0 | 19 | 2 | 12 | 11 | 8 | 5 | 1 | 7 | 9 | 20 | 3 | 14 | 4 | 13 | 16 | 15 | 10 | 6 | 17 | 18 | 0 |
| | | | | 0 | 0 | 0 | 0 | 0 | 0 | 0 | 0 | 0 | 0 | 0 | 0 | 0 | 0 | 0 | 0 | 0 | 0 | 0 | 0 | |
| 4.75 | 14.1 | 2352.6 | 933.0 | 5 | 11 | 2 | 19 | 20 | 12 | 8 | 3 | 7 | 9 | 4 | 1 | 10 | 14 | 13 | 15 | 17 | 18 | 16 | 6 | 0 |
| | | | | 0 | 0 | 0 | 0 | 0 | 0 | 0 | 0 | 0 | 0 | 0 | 0 | 0 | 0 | 0 | 0 | 0 | 0 | 0 | 0 | |
| 5 | 13.8 | 2408.3 | 933.0 | 19 | 5 | 2 | 11 | 12 | 1 | 8 | 20 | 10 | 9 | 3 | 4 | 7 | 13 | 14 | 16 | 15 | 6 | 17 | 18 | 0 |
| | | | | 0 | 0 | 0 | 0 | 0 | 0 | 0 | 0 | 0 | 0 | 0 | 0 | 0 | 0 | 0 | 0 | 0 | 0 | 0 | 0 | |

**Table 5.** *Cont.*

| $C_5$ | SRT | TEL | ML | Recovery Scheme | | | | | | | | | | | | | | | | | | | | Number of Normal Modes |
|---|---|---|---|---|---|---|---|---|---|---|---|---|---|---|---|---|---|---|---|---|---|---|---|---|---|
| 5.25 | 14.4 | 2444.3 | 933.0 | 2 | 8 | 19 | 11 | 12 | 20 | 5 | 7 | 9 | 14 | 1 | 3 | 10 | 15 | 4 | 13 | 16 | 18 | 17 | 6 | 0 |
| | | | | 0 | 0 | 0 | 0 | 0 | 0 | 0 | 0 | 0 | 0 | 0 | 0 | 0 | 0 | 0 | 0 | 0 | 0 | 0 | 0 | |
| 5.5 | 13.8 | 2514.6 | 933.0 | 11 | 12 | 2 | 19 | 8 | 7 | 20 | 5 | 1 | 9 | 14 | 3 | 4 | 15 | 13 | 17 | 16 | 18 | 10 | 6 | 0 |
| | | | | 0 | 0 | 0 | 0 | 0 | 0 | 0 | 0 | 0 | 0 | 0 | 0 | 0 | 0 | 0 | 0 | 0 | 0 | 0 | 0 | |
| 5.75 | 14.3 | 2571.3 | 933.0 | 19 | 7 | 2 | 12 | 11 | 20 | 8 | 5 | 1 | 9 | 3 | 4 | 15 | 14 | 10 | 13 | 16 | 17 | 6 | 18 | 0 |
| | | | | 0 | 0 | 0 | 0 | 0 | 0 | 0 | 0 | 0 | 0 | 0 | 0 | 0 | 0 | 0 | 0 | 0 | 0 | 0 | 0 | |
| 6 | 13.3 | 2629.1 | 933.0 | 19 | 5 | 8 | 9 | 2 | 12 | 14 | 20 | 1 | 7 | 3 | 11 | 15 | 4 | 16 | 13 | 10 | 18 | 17 | 6 | 0 |
| | | | | 0 | 0 | 0 | 0 | 0 | 0 | 0 | 0 | 0 | 0 | 0 | 0 | 0 | 0 | 0 | 0 | 0 | 0 | 0 | 0 | |

TEL: total economic loss; ML: maintenance loss.

From Table 5, it is difficult for us to analyze the relationship between the recovery sequence and the completion time of each damaged bridge. To better analyze the restoration process, we use the decoding method to obtain information on the restoration sequence of damaged bridges, the completion time of each bridge, and the task assignment of engineering tasks. The matrix in Table 6 shows all the mentioned information. The first row is the restoration sequence; the second row is the completion time of the corresponding bridge in the first row; and the third row shows which engineering team is used to repair the bridge in the first row. From Table 6, the percentage of repaired bridges in the first sequence can be evaluated, and the repair sequence of the damaged bridges can be summarized as follows. Bridge 30 is the first restoration bridge in the recovery scheme, accounting for 95.83%, while Bridge 12 accounts for 4.17%. In this way, we can determine that Bridge 30 (100%), Bridge 12 (91.67%), and Bridge 2 (87.5%) should be considered as priorities when we need to repair the first four damaged bridges. If we know the possible $y$ elements in the first $Y$ positions, the number of possible recovery schemes will be reduced to $y!(n-Y)!/n!$ of the original possible recovery schemes, which will reduce the running time sharply. Moreover, considering the recovery time of the bridges, the number of repaired bridges decreases with $C_5$ within the required period. For example, we can compare the number of repaired bridges within the same period. There are 7 bridges that can be restored when $C_5 = 0.25$, while there are almost 14 bridges that can be restored when $C_5 > 0.5$. Therefore, increasing $C_5$ reasonably can speed up the restoration process to reduce the indirect loss. Therefore, the summarized restoration sequence can reduce the running time, and increasing $C_5$ reasonably can promote engineering teams to complete the repair tasks quickly.

**Table 6.** Changes in the restoration sequence as the $C_5$ increases.

| $C_5$ | $B_0$, $T_0$, and Engineering Teams | | | | | | | | | | | | | | | | | | | |
|---|---|---|---|---|---|---|---|---|---|---|---|---|---|---|---|---|---|---|---|---|
| 0.25 | 30 | 12 | 5 | 9 | 28 | 24 | 22 | 27 | 16 | 33 | 21 | 2 | 19 | 17 | 26 | 14 | 11 | 1 | 3 | 10 |
| | 0.1 | 0.6 | 4.05 | 4.28 | 6.2 | 6.4 | 6.99 | 8.49 | 8.94 | 9.64 | 9.76 | 10.41 | 10.76 | 11.65 | 11.79 | 12.23 | 13.88 | 16.54 | 16.57 | 19.44 |
| | 2 | 4 | 4 | 3 | 2 | 2 | 4 | 2 | 2 | 2 | 1 | 1 | 1 | 2 | 3 | 1 | 3 | 1 | 4 | 2 |
| 0.5 | 30 | 12 | 2 | 33 | 9 | 16 | 17 | 27 | 22 | 14 | 24 | 11 | 1 | 5 | 3 | 21 | 19 | 28 | 26 | 10 |
| | 0.53 | 0.6 | 0.65 | 1.23 | 2 | 2 | 4.01 | 4.09 | 4.17 | 4.49 | 4.69 | 5.27 | 6.31 | 8.72 | 8.78 | 10.41 | 10.76 | 12.41 | 16.23 | 16.57 |
| | 4 | 2 | 3 | 4 | 2 | 1 | 2 | 1 | 4 | 1 | 1 | 4 | 1 | 2 | 3 | 3 | 4 | 1 | 2 | |
| 0.75 | 30 | 12 | 11 | 9 | 16 | 22 | 24 | 27 | 17 | 33 | 14 | 28 | 21 | 2 | 19 | 3 | 1 | 5 | 26 | 10 |
| | 0.1 | 0.6 | 0.68 | 2 | 2.08 | 3.54 | 3.74 | 4.17 | 4.7 | 4.87 | 6.17 | 8.1 | 8.33 | 8.41 | 8.45 | 9.64 | 11.78 | 11.86 | 15.84 | 16.24 |
| | 4 | 2 | 4 | 1 | 4 | 2 | 2 | 4 | 3 | 1 | 2 | 3 | 1 | 4 | 4 | 3 | 2 | 1 | | |
| 1 | 30 | 12 | 2 | 33 | 16 | 9 | 1 | 17 | 11 | 14 | 21 | 19 | 3 | 5 | 27 | 22 | 24 | 28 | 10 | 26 |
| | 0.1 | 0.6 | 0.65 | 1.3 | 1.5 | 2 | 3.44 | 3.51 | 4.09 | 4.49 | 5.24 | 6.06 | 6.77 | 6.89 | 8.15 | 8.18 | 8.35 | 12.99 | 14.56 | 15.69 |
| | 4 | 2 | 3 | 2 | 4 | 1 | 2 | 4 | 4 | 4 | 3 | 4 | 1 | 2 | 4 | 3 | 4 | 2 | 1 | 3 |
| 1.25 | 30 | 12 | 2 | 9 | 14 | 17 | 11 | 24 | 1 | 22 | 16 | 33 | 19 | 21 | 27 | 3 | 5 | 28 | 26 | 10 |
| | 0.1 | 0.6 | 0.65 | 2 | 2.07 | 2.11 | 4.16 | 4.94 | 4.96 | 5.05 | 5.5 | 6.2 | 6.55 | 6.59 | 7.03 | 9.73 | 10 | 12.69 | 14.54 | 17.52 |
| | 4 | 2 | 3 | 1 | 2 | 4 | 2 | 2 | 3 | 4 | 4 | 4 | 1 | 2 | 3 | 4 | 1 | 2 | 3 | |
| 1.5 | 30 | 12 | 2 | 33 | 9 | 17 | 14 | 21 | 19 | 5 | 3 | 16 | 27 | 26 | 28 | 10 | | | | |
| | 0.1 | 0.6 | 0.65 | 0.8 | 2 | 2.79 | 2.81 | 3.21 | 5.19 | 5.54 | 5.73 | 5.74 | 6.32 | 6.66 | 6.77 | 6.77 | 8.75 | 10.07 | 12.87 | 13.52 |
| | 4 | 2 | 3 | 4 | 1 | 3 | 4 | 4 | 2 | 2 | 3 | 2 | 4 | 2 | 1 | 4 | 1 | 2 | 3 | |
| 1.75 | 30 | 12 | 33 | 2 | 9 | 17 | 14 | 1 | 11 | 19 | 21 | 16 | 3 | 5 | 26 | 22 | 24 | 27 | 10 | 28 |
| | 0.1 | 0.6 | 0.8 | 1.25 | 2 | 2.81 | 3.21 | 3.39 | 3.79 | 4.14 | 4.59 | 4.59 | 6.77 | 6.84 | 7.89 | 9.71 | 9.91 | 11.85 | 12.38 | 13.99 |
| | 4 | 2 | 4 | 2 | 1 | 4 | 4 | 2 | 4 | 4 | 3 | 1 | 2 | 3 | 1 | 4 | 2 | 4 | 3 | |
| 2 | 30 | 12 | 2 | 11 | 33 | 9 | 16 | 1 | 17 | 3 | 5 | 24 | 22 | 14 | 27 | 21 | 19 | 26 | 10 | 28 |
| | 0.1 | 0.6 | 0.65 | 1.18 | 1.88 | 2 | 2.33 | 2.79 | 4.34 | 4.87 | 5.45 | 5.65 | 6.05 | 7.82 | 8.93 | 9.35 | 12.66 | 13.92 | | |
| | 4 | 2 | 1 | 2 | 2 | 3 | 2 | 1 | 2 | 4 | 3 | 3 | 1 | 3 | 1 | 2 | 2 | 3 | 4 | 1 |
| 2.25 | 30 | 12 | 2 | 9 | 17 | 14 | 1 | 16 | 5 | 3 | 24 | 22 | 27 | 21 | 19 | 26 | 10 | 28 | | |
| | 0.1 | 0.6 | 0.7 | 1.25 | 1.83 | 2 | 2.11 | 2.51 | 3.97 | 4.42 | 5.45 | 5.47 | 7.04 | 7.1 | 7.3 | 7.36 | 9.39 | 10.34 | 13.24 | 13.46 |
| | 4 | 2 | 3 | 2 | 1 | 4 | 4 | 2 | 1 | 3 | 3 | 4 | 4 | 2 | 4 | 3 | 1 | 4 | | |
| 2.5 | 30 | 2 | 33 | 12 | 17 | 1 | 14 | 9 | 16 | 11 | 26 | 5 | 3 | 19 | 22 | 24 | 21 | 27 | 28 | 10 |
| | 0.1 | 0.65 | 0.7 | 1.3 | 2.01 | 2.24 | 2.64 | 2.65 | 3.1 | 3.22 | 4.6 | 6.67 | 6.78 | 7.02 | 7.54 | 7.56 | 7.69 | 9.63 | 13.66 | 14.81 |
| | 4 | 3 | 2 | 2 | 1 | 3 | 3 | 4 | 2 | 4 | 1 | 4 | 2 | 1 | 3 | 2 | 4 | | | |
| 2.75 | 30 | 11 | 2 | 12 | 9 | 17 | 22 | 24 | 14 | 19 | 21 | 33 | 16 | 1 | 3 | 5 | 27 | 26 | 28 | 10 |
| | 0.1 | 0.58 | 0.75 | 1.35 | 2 | 3.36 | 3.52 | 3.72 | 4.12 | 4.47 | 4.59 | 5.29 | 5.74 | 6.61 | 6.77 | 6.81 | 7.83 | 10.07 | 12.91 | 14.4 |
| | 3 | 2 | 3 | 3 | 1 | 3 | 2 | 2 | 2 | 2 | 4 | 4 | 4 | 2 | 1 | 3 | 4 | 1 | 3 | 2 |

**Table 6.** *Cont.*

| $c_5$ | $B_0$, $T_0$, and Engineering Teams | | | | | | | | | | | | | | | | | | | |
|---|---|---|---|---|---|---|---|---|---|---|---|---|---|---|---|---|---|---|---|---|
| 3 | 30 | 12 | 2 | 33 | 9 | 17 | 14 | 11 | 1 | 22 | 24 | 16 | 26 | 21 | 19 | 3 | 5 | 27 | 28 | 10 |
|  | 0.1 | 0.6 | 0.65 | 1.3 | 2 | 2.11 | 2.4 | 2.69 | 3.44 | 3.59 | 3.64 | 4.04 | 6.94 | 6.99 | 7.29 | 7.46 | 7.49 | 9.55 | 13.09 | 15.08 |
|  | 4 | 2 | 3 | 2 | 1 | 4 | 1 | 4 | 2 | 3 | 2 | 3 | 2 | 1 | 2 | 4 | 3 | 4 | 1 | 2 |
| 3.25 | 30 | 12 | 2 | 33 | 9 | 17 | 1 | 14 | 11 | 3 | 5 | 19 | 24 | 22 | 16 | 27 | 21 | 26 | 10 | 28 |
|  | 0.1 | 0.6 | 0.65 | 0.8 | 2 | 2.66 | 2.94 | 3.06 | 3.52 | 5.37 | 5.45 | 5.8 | 6 | 6 | 6.45 | 8.09 | 8.11 | 9.75 | 13.16 | 14.19 |
|  | 4 | 2 | 3 | 4 | 1 | 3 | 4 | 3 | 4 | 2 | 1 | 1 | 3 | 1 | 1 | 3 | 4 | 1 | 2 | 3 |
| 3.5 | 30 | 12 | 2 | 19 | 33 | 9 | 17 | 14 | 1 | 16 | 11 | 22 | 24 | 21 | 3 | 5 | 27 | 26 | 10 | 28 |
|  | 0.1 | 0.6 | 0.65 | 1 | 1.7 | 2 | 2.11 | 2.51 | 2.74 | 2.96 | 3.32 | 5.9 | 6.1 | 6.29 | 6.77 | 6.77 | 8.86 | 9.59 | 10.24 | 12.2 |
|  | 4 | 2 | 3 | 3 | 3 | 1 | 4 | 4 | 2 | 4 | 2 | 4 | 4 | 3 | 1 | 2 | 2 | 3 | 1 | 4 |
| 3.75 | 12 | 33 | 30 | 2 | 9 | 11 | 1 | 22 | 24 | 16 | 17 | 19 | 14 | 3 | 5 | 21 | 26 | 27 | 28 | 10 |
|  | 0.6 | 0.7 | 0.8 | 1.45 | 2 | 2.03 | 2.74 | 2.94 | 2.94 | 3.39 | 4.01 | 4.36 | 4.76 | 6.8 | 6.84 | 7.53 | 8.06 | 8.89 | 13.63 | 14.63 |
|  | 2 | 4 | 4 | 4 | 1 | 4 | 2 | 3 | 2 | 3 | 1 | 1 | 1 | 4 | 3 | 2 | 1 | 4 | 2 | 3 |
| 4 | 30 | 12 | 2 | 33 | 19 | 16 | 11 | 9 | 1 | 17 | 14 | 22 | 5 | 21 | 3 | 24 | 27 | 26 | 10 | 28 |
|  | 0.1 | 0.6 | 0.65 | 0.8 | 1 | 1.25 | 1.83 | 2 | 2.74 | 3.01 | 3.14 | 6.08 | 6.46 | 6.59 | 6.6 | 6.66 | 8.69 | 9.38 | 10.13 | 12.69 |
|  | 4 | 2 | 3 | 4 | 3 | 4 | 4 | 1 | 2 | 3 | 2 | 2 | 3 | 1 | 4 | 3 | 4 | 2 | 3 | 1 |
| 4.25 | 30 | 19 | 12 | 33 | 2 | 14 | 17 | 16 | 9 | 1 | 22 | 24 | 3 | 5 | 21 | 27 | 26 | 10 | 28 | |
|  | 0.1 | 0.35 | 0.6 | 0.8 | 1 | 1.38 | 1.78 | 2.01 | 2.46 | 2.6 | 3.14 | 5.4 | 5.6 | 6.55 | 6.59 | 7.19 | 7.69 | 9.85 | 10.06 | 13.29 |
|  | 4 | 1 | 2 | 4 | 1 | 4 | 3 | 3 | 2 | 1 | 3 | 3 | 4 | 1 | 2 | 3 | 4 | 1 | 2 | |
| 4.5 | 30 | 19 | 2 | 12 | 11 | 14 | 17 | 9 | 33 | 1 | 22 | 5 | 24 | 16 | 3 | 21 | 26 | 27 | 10 | 28 |
|  | 0.1 | 0.35 | 0.65 | 0.7 | 1.28 | 1.68 | 2.01 | 2.35 | 2.38 | 2.79 | 5.29 | 5.83 | 6.03 | 6.48 | 6.78 | 7.38 | 8.59 | 8.87 | 9.95 | 13.48 |
|  | 1 | 3 | 2 | 1 | 1 | 1 | 4 | 3 | 1 | 2 | 3 | 1 | 1 | 4 | 2 | 3 | 4 | 1 | 1 | 2 |
| 4.75 | 30 | 2 | 33 | 19 | 12 | 11 | 9 | 17 | 14 | 16 | 1 | 5 | 24 | 3 | 22 | 27 | 21 | 26 | 10 | 28 |
|  | 0.1 | 0.65 | 0.8 | 1 | 1.4 | 1.98 | 2 | 2.01 | 2.38 | 2.83 | 4.15 | 5.45 | 5.65 | 5.77 | 5.77 | 7.74 | 8.74 | 9.07 | 11.21 | 11.87 |
|  | 4 | 3 | 4 | 3 | 4 | 4 | 1 | 2 | 4 | 4 | 2 | 1 | 1 | 4 | 3 | 1 | 2 | 4 | 1 | 3 |
| 5 | 30 | 19 | 2 | 12 | 9 | 17 | 14 | 16 | 1 | 11 | 5 | 22 | 24 | 3 | 21 | 27 | 26 | 10 | 28 | |
|  | 0.1 | 0.45 | 0.65 | 1.25 | 1.95 | 2 | 2.01 | 2.4 | 2.4 | 2.59 | 2.98 | 5.85 | 5.92 | 6.12 | 6.78 | 7.18 | 8.87 | 9.15 | 9.59 | 13.28 |
|  | 1 | 1 | 3 | 3 | 2 | 4 | 3 | 2 | 1 | 3 | 2 | 3 | 3 | 4 | 1 | 4 | 2 | 3 | 1 | |
| 5.25 | 30 | 19 | 12 | 2 | 33 | 11 | 14 | 17 | 9 | 16 | 24 | 1 | 22 | 5 | 3 | 26 | 21 | 27 | 10 | 28 |
|  | 0.1 | 0.45 | 0.6 | 0.65 | 1.15 | 1.23 | 1.55 | 2.01 | 2.6 | 3.05 | 3.25 | 3.69 | 4.17 | 6.7 | 6.78 | 7.47 | 8.28 | 8.87 | 10.94 | 12.8 |
|  | 3 | 3 | 2 | 1 | 3 | 1 | 4 | 2 | 2 | 2 | 3 | 1 | 2 | 4 | 1 | 3 | 4 | 1 | 4 | 2 |
| 5.5 | 30 | 19 | 2 | 12 | 11 | 33 | 14 | 17 | 9 | 1 | 24 | 22 | 5 | 3 | 27 | 16 | 21 | 26 | 10 | 28 |
|  | 0.1 | 0.35 | 0.65 | 0.7 | 0.93 | 1.35 | 1.75 | 2.01 | 2.7 | 3.07 | 3.27 | 4.69 | 6.15 | 6.78 | 6.78 | 7.23 | 7.86 | 9.45 | 10.7 | 12.88 |
|  | 4 | 2 | 3 | 4 | 2 | 3 | 3 | 1 | 4 | 2 | 2 | 3 | 4 | 3 | 1 | 3 | 2 | 4 | 3 | 1 |
| 5.75 | 30 | 19 | 11 | 2 | 33 | 12 | 14 | 17 | 9 | 24 | 1 | 16 | 5 | 22 | 3 | 27 | 21 | 26 | 10 | 28 |
|  | 0.1 | 0.35 | 0.58 | 0.65 | 1.05 | 1.18 | 1.58 | 2.11 | 2.65 | 2.85 | 3.19 | 3.64 | 5.56 | 5.79 | 6.35 | 7.88 | 8.23 | 8.86 | 9.82 | 13.98 |
|  | 1 | 4 | 2 | 3 | 4 | 2 | 2 | 1 | 3 | 4 | 4 | 1 | 3 | 2 | 3 | 4 | 1 | 2 | 3 | |
| 6 | 30 | 14 | 12 | 2 | 19 | 33 | 9 | 11 | 1 | 24 | 22 | 17 | 5 | 3 | 26 | 16 | 21 | 27 | 10 | 28 |
|  | 0.1 | 0.4 | 0.6 | 0.75 | 0.75 | 1.45 | 2 | 2.03 | 2.89 | 3.09 | 3.54 | 4.04 | 6.54 | 6.77 | 6.84 | 6.99 | 8.63 | 8.93 | 10.46 | 12.87 |
|  | 1 | 4 | 3 | 4 | 1 | 1 | 2 | 1 | 4 | 4 | 3 | 1 | 4 | 2 | 3 | 4 | 1 | 3 | 4 | 2 |

### 4.3. Discussion

Through analyzing the results of the above simulation experiments, some viewpoints occur, as follows.

- Indirect loss increases with the increase in damaged bridges and decreases as engineering teams increase. Once the damaged bridges are determined, arranging more engineering teams can reduce the indirect loss effectively.
- A cost-effective management of engineering teams can reduce team management loss, and we can also consider using more teams to restore the network for decreasing indirect loss.
- The higher price of emergency service promotes all teams to use the expedited mode for restoring damaged bridges as soon as possible, and bridges with shorter maintenance time should be repaired as a priority.
- To minimize total economic loss, it is essential to design the optimal recovery scheme (repair sequence and repair mode) wisely to balance the conflicts between indirect loss and direct loss.

### 5. Conclusions

This paper analyzes economic loss comprehensively by considering maintenance loss, team management loss, and indirect economic loss caused as a result of energy consumption and emergency service. The GA is used to solve the complex optimization problem to obtain the optimal recovery scheme (repair sequence and repair modes). Some important findings are summarized as follows. (1) Indirect loss accounts for about half of economic loss, while the higher price of emergency service promotes a reduction in the indirect loss using the expedited modes to repair damaged bridges as soon as possible. (2) Direct loss increases with the increase in engineering teams because of the sharp increase in management loss, while more engineering teams can decrease the indirect loss. Thus, finding an optimal number of engineering teams is important to balance the conflict between indirect loss and management loss.

However, this paper also has some limitations that need to be solved in future research. (1) The research work pays more attention to the recovery optimization problem under the known risks without considering the influence of uncertain risks. (2) Also, for the

indirect loss, this paper did not consider the loss caused by the cost of time. Therefore, in the future, it will be essential to consider recovery optimization in terms of uncertain information, including recovery time and disaster risks. Moreover, the economic loss can also be improved by further considering the time factor.

**Author Contributions:** Conceptualization: J.Z.; Methodology: M.L.; Software: M.L. and Z.Z.; Validation, M.L. and Q.L.; Formal analysis: M.L. and Z.Z.; Investigation: M.L.; Resources: Z.Z. and X.C.; Data curation: M.L.; Writing—original draft preparation: M.L. and Z.Z.; Writing—review and editing: J.Z. and X.C.; Supervision: Z.C. and J.Z.; Project administration: Z.C.; Funding acquisition: J.Z. and Q.L. All authors have read and agreed to the published version of the manuscript.

**Funding:** This research was supported by the National Natural Science Foundation of China under grant numbers 72101202, 52005400; China Postdoctoral Science Foundation under grant number 2022MD713793; and the Outstanding Youth Science Fund of Xi'an University of Science and Technology under grant number 22002.

**Institutional Review Board Statement:** Not applicable.

**Informed Consent Statement:** Not applicable.

**Data Availability Statement:** Data is contained within the article.

**Conflicts of Interest:** The authors declare no conflict of interest.

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
