# Peer review of "Post-Disaster Resilience Optimization for Road–Bridge Transportation Systems Considering Economic Loss"

_sustainability, doi:10.3390/su151914380_

Round 1

Reviewer 1 Report

1. The literature review part is not thorough and only lists some existing research. A comprehensive overview and literature reorganization is necessary.

2. p.2 line 82. Isn't ‘indirect loss caused by emergency service’ a special social economic category?

3. What’s post-disaster? It is recommended to give a definition of post-disaster in section 1

4. p3. Line 99. Why did the author call the road-bridge transportation system an independent system?

5. In general, both damage level and recovery time are stochastic rather than fixed. Can the optimization mode as well as the solving algorithm extend to the random case?

6. p3. Line 127. Please give an explanation on ‘… the engineering team will select the bridge with the lower recovery time when this team is idle’.

7. p4. Please check Eq. (3).

8. Why does the author select GA rather than other heuristic algorithms?

9. What’s the relationship between the ‘Recovery sequence 6 3 2 4 1 5’ (in figure 2) and B0={3 6 4 2 5 1} (p7.259)?

10. There are some problems in sentence structures (e.g. p.10 line 317) and first letter capitalization (e.g. p.17 line 436).

Author Response

Dear reviewer,

Thank you so much for your review and advice on our research work.

We have revised the paper based on the reviewer’s comments and suggestions point-to-point. The revised parts have been marked with red color in the revised manuscript. We would like to submit the revised manuscript to Sustainability for further review and to be considered for publication.

Our response to the comments and suggestions, please see the attachment. We hope you find the revisions acceptable.

Yours sincerely,

Jiangbin Zhao

Reviewer 2 Report

The introduction needs more work to clearly establish the gap and novelty of this study. The authors state this paper considers total economic loss including indirect costs, but many of the cited papers also consider indirect economic impacts. The authors should clearly explain how their approach to calculating total economic loss is more comprehensive than previous studies.

The assumptions need to be stated more clearly. For example, it is not clear what information is known after the disaster (e.g. damage level of each bridge). Also, how are the recovery times and costs for the two repair modes determined? This needs to be explained.

The case study should provide more details on the data used for the experiments, such as how the bridge damage levels, recovery times, costs etc. were determined. The validity of the case study depends on realistic data.

The results and discussion focus heavily on costs but lack analysis of the recovery schedules and times. As recovery time is a key aspect of resilience, this should be analyzed in more depth.

The conclusions are quite brief. More emphasis could be placed on the practical implications of the findings, limitations, and future research needs.

The introduction needs more work to clearly establish the gap and novelty of this study. The authors state this paper considers total economic loss including indirect costs, but many of the cited papers also consider indirect economic impacts. The authors should clearly explain how their approach to calculating total economic loss is more comprehensive than previous studies.

The assumptions need to be stated more clearly. For example, it is not clear what information is known after the disaster (e.g. damage level of each bridge). Also, how are the recovery times and costs for the two repair modes determined? This needs to be explained.

The case study should provide more details on the data used for the experiments, such as how the bridge damage levels, recovery times, costs etc. were determined. The validity of the case study depends on realistic data.

The results and discussion focus heavily on costs but lack analysis of the recovery schedules and times. As recovery time is a key aspect of resilience, this should be analyzed in more depth.

The conclusions are quite brief. More emphasis could be placed on the practical implications of the findings, limitations, and future research needs.

Author Response

(The authors gave the same response as above.)

Reviewer 3 Report

This paper presents an optimization model to determine the recovery strategy for road-bridge transportation systems after a disaster. Though the study might be useful, i found it difficult to understand the method. My specific comments are as follows:

The notation in equation 1-12 need to be revised to improve clarity. For example, several notations are used long before they are defined.

I believe that there are typos in equation (3). For example, why variable x is multiplied by index j?

Equation (5),  P(b,t)= 0 if there is more than one shortest pathway between two nodes of the bridge b_i. My understanding of this setting is that the model assumes that there will not be additional fuel cost if there is an alternative shortest path. However, what if the alternative shortest pathways also have damaged bridges (bridges other than b_i)? Why the duration of the additional energy cost is t_i - t_(i-1)? Shouldn’t the energy cost exist after bridge b_i is repaired (i.e., the duration should be t_i). The cost calculation does not seem to be correct either. If C_4 is the unit price per vehicle traveling one km, then what is the meaning of L_(I_1)?

The same issues with equation (6).

The authors consider the economic loss of the energy cost of travelers. Could the time cost of travelers be considered as well?

The English writing may need moderate editing.

Author Response

(The authors gave the same response as above.)

Round 2

Reviewer 1 Report

The author has revised the manuscript according to the comments. I recommend to accept the manuscript in its present form.

Reviewer 3 Report

The authors have improved their paper and made significant efforts to address my concerns. I have no more comments.